# MAPT Mutations V337M and N297K Alter Organelle Trafficking in Frontotemporal Dementia Patient-Specific Motor Neurons

**DOI:** 10.3390/biomedicines12030641

**Published:** 2024-03-13

**Authors:** Christiane Hartmann, Marie Anskat, Marc Ehrlich, Jared Sterneckert, Arun Pal, Andreas Hermann

**Affiliations:** 1Translational Neurodegeneration Section “Albrecht Kossel”, Department of Neurology, University Medical Center Rostock, 18057 Rostock, Germany; christiane.hartmann@med.uni-rostock.de; 2Division for Neurodegenerative Diseases, Department of Neurology, Technische Universität Dresden, 01307 Dresden, Germany; marie.anskat@gmx.de (M.A.); a.pal@hzdr.de (A.P.); 3Novartis Pharma GmbH, 90429 Nuernberg, Germany; marc.ehrlich@novartis.com; 4Center for Regenerative Therapies TU Dresden (CRTD), Technische Universität Dresden, 01307 Dresden, Germany; jared.sterneckert@tu-dresden.de; 5Medical Faculty Carl Gustav Carus, Technische Universität Dresden, 01069 Dresden, Germany; 6Dresden High Magnetic Field Laboratory (HLD-EMFL), Helmholtz-Zentrum Dresden-Rossendorf (HZDR), 01328 Dresden, Germany; 7Center for Transdisciplinary Neurosciences (CTNR), University Medical Center Rostock, 18057 Rostock, Germany; 8German Center for Neurodegenerative Diseases (DZNE), Rostock/Greifswald, 18147 Rostock, Germany

**Keywords:** FTD, MAPT, TAU, organelle trafficking, mitochondria, lysosome, axonal outgrowth, microtubule, tubulin modification

## Abstract

Frontotemporal dementia (FTD) is a neurodegenerative disease characterized by the progressive loss of neurons mainly in the frontal and temporal lobes of the brain. Mutations (e.g., V337M, N297K) in the microtubule-associated protein TAU (MAPT) are responsible 5–20% of familial FTD cases and have been associated with defects in organelle trafficking that plays a critical role in the proper function of cells, including transport of essential molecules and degradation of waste products. Due to the critical role of TAU mutations in microtubule stabilization and organelle transportation, it is of great interest to study these molecular mechanisms to develop effective therapeutic strategies. Therefore, herein, we analyzed mitochondrial and lysosomal trafficking in disease-specific spinal motor neurons by using live cell imaging in undirected (uncompartmentalized) and directed (compartmentalized) cell culture systems. While V337M neurons only expressed 3R TAU, the N297K mutant neurons expressed both 3R and 4R TAU. Axonal trafficking was affected differentially in V337M and N297 MAPT mutated neurons. These findings suggest that the MAPT mutations V337M and N297K impaired axon physiology differentially, which highlights the need for mutation- and/or 3R/4R TAU-specific therapeutic approaches.

## 1. Introduction

Frontotemporal dementia (FTD) is a group of neurodegenerative disorders that primarily affect the frontal and temporal lobes of the brain leading to changes in behavior, personality and language [1,2]. As we face an aging society in the future, understanding and developing new therapies for age-related diseases will become increasingly important. It is the second most common cause of dementia in individuals under the age of 60, and accounts for approximately 10% of all dementia cases [3]. Most studies indicate that men and women are affected equally often [4,5]. Furthermore, 25–45% of FTD patients have a positive family history, with mutations in the genes for the open reading frame 72 of chromosome 9 (C9orf72), progranulin (GRN), or the MAPT each accounting for around 10–20% of patients [6]. The pathology of FTD is characterized by the presence of abnormal protein aggregates in affected neurons that disrupt healthy cellular functions, leading to loss of neurons and their connections [1,2,3,7]. Approximately 30% of patients have inherited forms of FTD, and 5–20% of these patients carry mutations in the MAPT gene on chromosome 17q21, which encodes the microtubule-associated protein TAU [8,9]. FTD patients with MAPT mutations exhibit dementia and Parkinson-like motor impairment due to additional degeneration in subcortical brain areas including the substantia nigra. Therefore, this form has been termed FTD and Parkinsonism associated with chromosome 17 (FTDP-17) [8,10].

TAU is predominantly expressed in neurons and to a lower extent in glial cells. This protein is important for stabilization of the cell cytoskeleton due to microtubule binding as well as axoplasmic transport [11,12]. MAPT mutations lead to the accumulation of misfolded TAU protein. Clinically, these heterogeneous diseases display phenotypes including cognitive and behavioral, language and movement disturbances as well as non-specific amnestic symptoms with increasing age [13]. There are primary tauopathies, where TAU plays the major role in pathology (FTDs), and secondary tauopathies, where TAU aggregation is considered to be a consequence of other pathological events (amyloid beta in Alzheimer’s disease, repetitive brain injury in chronic traumatic encephalopathy, etc.) [13,14]. As of now, more than 26 different tauopathies have been identified [14]. The pathological diversity of tauopathies can be explained, at least in part, by the different isoforms and post-translational modifications of TAU. Alternative splicing and phosphorylation of TAU are cellular mechanisms that regulate microtubule stability. Six isoforms of TAU are expressed in the adult brain produced by alternative splicing of exons 2, 3 and 10 [8] of MAPT. Regulated inclusion of exons 2 and 3 results in TAU isoforms with 0, 1, or 2 N-terminal inserts, while exclusion or inclusion of exon 10 results in the expression of TAU isoforms with three (3R) or four (4R) c-terminally microtubule-binding repeats. In the healthy brain, the ratio of 3R and 4R is approximately 1:1, whereby this ratio can be substantially altered during disease. For example, in FTD an increased ratio of the 4R to 3R TAU is described [15]. Depending on exon 10 splicing, tauopathies can be categorized into 3R (e.g., Pick’s disease, FTD), 4R (e.g., supranuclear palsy) and 3/4R (e.g., Alzheimer’s disease) based on the dominant isoforms found in TAU aggregates [16]. Isoforms with 4R are more strongly associated with microtubules than 3R ones [17]. Some FTDP-17 patients display a ratio shift towards 4R isoforms, leading to an increase in deposition of 4R TAU within neurons and glial cells [8]. The altered ratio of 3R/4R primes a disrupted balance of microtubules and dissociation of TAU. Free TAU interacts with filamentous actin and aggregates in dendrites and dendritic spines. These aggregations are caused by conformational changes and can form fibrils in neurons and glial cells as well. TAU fibrils were shown to have prion-like properties and spread via synapse connection to distal regions of the brain where they can lead to brain atrophy, neuronal death, and dementia [17,18,19,20,21]. Another pathological characterization of tauopathies is an abnormal hyperphosphorylation [22] that alters the binding of TAU to microtubules as well. The phosphorylation of TAU determines the amount of TAU that binds to microtubules, meaning increased phosphorylation reduces the amount of microtubules binding TAU [23].The relationship between MAPT splicing, TAU phosphorylation, and TAU aggregation is still not fully understood.

The two specific mutations, V337M MAPT and N279K MAPT show a particularly high prevalence in FTDP-17 patients. The V337M MAPT mutation results in a substitution of valine (V) for methionine (M) at position 337 of TAU. This mutation is located in the microtubule-binding domain of TAU, which is critical for the proper function of microtubules in neurons, especially concerning neuronal plasticity and axonal outgrowth [11,24]. V337M mutation was shown to promote self-aggregation and reduce binding to microtubules [25,26,27,28,29]. The N279K MAPT mutation results in a substitution of lysine (K) for asparagine (N) at position 279 of TAU. This mutation is located on exon 10, which is important for regulating the function and localization of TAU in neurons. Studies have shown that the N279K mutation results in increased TAU aggregation and altered TAU localization in neurons, leading to formation of neurofibrillary tangles, a hallmark of FTD. N279K mutation was shown to shift the 3R/4R ratio by increasing 4R isoforms through a strengthened enhancer in exon 10 and following increased inclusion [30,31,32]. Neurons derived from induced pluripotent stem cells (iPSCs) from individuals with FTDP-17-associated MAPT mutation V337M or N279K displayed marked TAU pathology with an increase in fragmentation and phospho-TAU immunoreactivity and decreased neurite extension [33]. Ehrlich and colleagues further demonstrated activation of unfolded protein response in FTDP-17 neurons as well as FTDP-17-associated gene expression profiles. Furthermore, research has uncovered a link between TAU mutations and defects in organelle trafficking, which is essential for proper cellular function [34,35,36].

Organelle trafficking refers to the process by which different types of organelles, such as mitochondria and lysosomes, are transported within neurons to maintain their proper function [37,38,39]. Defects in organelle trafficking can alter cellular clearance pathways leading to protein aggregation and can cause neurodegeneration [40]. Mitochondria are important cell organelles for a variety of processes like bioenergetics, intermediate metabolism, intrinsic cell death, apoptosis induction, and calcium homeostasis [41]. The majority of mitochondria are transported based on microtubules and actin filaments [42,43,44]. To maintain these essential activities and energy homeostasis, mitochondria need to be precisely distributed, sustained in the periphery, and cleared when necessary. Causes of neuronal degeneration could be consequently predicted by deregulated mitochondrial motility. Additionally, malfunctioning mitochondria are able to affect their motility through generated reactive oxygen species [45,46]. Indeed, both mutations, V337 and N297K, displayed increased oxidative stress when mitochondrial respiration was inhibited [33], which could be a hint for disrupted organelle trafficking. By contrast, lysosomes are responsible for the uptake and removal of dysfunctional or long-lived proteins and damaged/old organelles such as mitochondria. Due to frequently occurring protein misfolding, cells maintain a clearance mechanism (including lysosomal activity) called autophagy [47,48]. Dysfunctional autophagy can lead to protein aggregation or inclusions, which are common characteristics of neurodegenerative diseases. Damaged mitochondria preferentially fuse with lysosomes upon a region-specific lysosomal enrichment in the cell body [49].

Investigating the trafficking defects of mitochondria and lysosomes in FTDP-17 is of interest for two main reasons. First, TAU plays a critical role in microtubule stabilization and organelle transport, whereas TAU mutations are known to disrupt these processes, leading to organelle trafficking defects. These defects, in turn, contribute to the pathogenesis of FTD and other tauopathies [50]. Second, understanding the molecular mechanisms underlying organelle trafficking defects in FTD is essential for the development of effective therapeutic strategies [51]. Since TAU is mainly expressed in neuronal axons [52], we hypothesize mutations in MAPT will predominantly induce axonal phenotypes in iPSC-derived motor neurons. Therefore, our study focuses on the characterization of possible trafficking defects of mitochondria and lysosomes caused by TAU mutations V337M and N279K. Additionally, MAPT mutations have different effects on microtubule binding [29]. We thus hypothesized that V337M or N297K mutations will induce differential axonopathies.

For this, we used iPSC-derived motor neurons of FTDP-17 patients carrying the specific mutations V337M or N297K and analyzed them for their trafficking defects in comparison to healthy controls (WT1, WT2) in undirected (uncompartmentalized) and directed (compartmentalized) cell cultures using live cell imaging as well as immunofluorescence staining. The latter served also to selectively image the axons. Mitochondrial and lysosomal mean speed and organelle displacement as well as mitochondrial membrane potential were measured by live cell imaging, which allowed the analysis of possible membrane potential loss and consequently lack of energy in the motor neurons’ axons.

## 2. Materials and Methods

### 2.1. Differentiation of Human Neuronal Progenitor Cells (NPCs) to Spinal Motor Neurons (sMNs)

The derivation of human NPCs from fibroblasts was already reported [33]. Final sMN differentiation was induced as described in [53]. Briefly, NPCs were treated with 0.25 µg/mL PMA (Biomol, Hamburg, Germany), 3 µM CHIR99021 and 150 µM ascorbic acid in N2B27. After two days, 1 µM retinoic acid, 1 ng/mL BDNF and 1 ng/mL GDNF were added. On day 9, another split step was performed to seed the cells into the final cell culture format of choice (undirected or directed; see below). Furthermore, the medium was modified to induce neural maturation. For this purpose, the developing neurons were treated with N2B27 containing 10 ng/µL BDNF, 500 µM dbcAMP, 10 ng/µL GDNF150 µM ascorbic acid, and 1 ng/mL TGFβ3. Following this protocol, it was possible to keep cells in culture for over 2 months. If nothing else has been mentioned, supplements, media and solutions were from Thermo Fisher Scientific (Waltham, MA, USA). 

### 2.2. Microfluidic Chambers (MFCs)

The MFCs were purchased from Xona (RD900, Temecula, NC, USA). At first, glass-bottomed dishes (Merck, Darmstadt, Germany) with an inner diameter of 27 mm were coated with Poly-L-Ornithine (Merck, Darmstadt, Germany, P4957, 0.01% stock diluted 1:3 in PBS) overnight at 37 °C. After 3 steps of washing with sterile water, they were kept under the sterile hood for air-drying. MFCs were sterilized with 70% ethanol and left drying. Next, the MFCs were dropped onto the dishes and carefully pressed on the glass surface for firm adherence. The system was then perfused with Laminin (Roche, Basel, Switzerland 11243217001, 0.5 mg/mL stock diluted 1:50 in PBS) for 3 h at 37 °C. For seeding cells, the system was once washed with medium, and then 10 µL containing a high concentration of cells (3 × 10^7^ cells/mL) was directly injected into the main channel connecting two wells. After allowing for cell attachment over 30–60 min in the incubator, the still empty wells were filled up with maturation medium. This method had the advantage of increasing the density of neurons in direct juxtaposition to microchannel entries, whereas the wells remained cell-free, thereby reducing the medium turnover to a minimum. To avoid drying out, PBS was added around the MFCs. Two days after seeding, the medium was replaced in a manner that gave the neurons a guidance cue for growing through the micro-channels. Specifically, a growth factor gradient was established by adding 100 µL N2B27 with 500 µM dbcAMP to the proximal seeding site and 200 µL N2B27 with 500 µM dbcAMP, 10 ng/µL BNDF, 10 ng/µL GDNF and 100 ng/µL NGF to the distal exit site. The medium was replaced in this manner every third day. After 7 days, the first axons began spreading out at the exit site, and cells were typically maintained for up to six weeks.

### 2.3. Immunofluorescence Stainings

For immunofluorescence staining, cells were washed twice with PBS without Ca^2+^/Mg^2+^ and fixed with 4% PFA in PBS for 10 min at room temperature. PFA was aspirated off and cells were washed three times with PBS. Fixed cells were first permeabilized for 10 min in 0.2% Triton X solution and subsequently incubated for 1 h at RT in blocking solution (1% BSA, 5% donkey serum, 0.3 M glycine and 0.02% Triton X in PBS). Following blocking, primary antibodies were diluted in blocking solution and cells were incubated with primary antibody solution overnight at 4 °C. The following primary antibodies were used: rabbit-anti-MAP2 (1:500, MERCK, Darmstadt, Germany #AB5622), chicken-anti-SMI32 (1:10,000, BioLegend, Koblenz, Germany, #PCK-592P), mouse-anti-TUJ-1 (1:700, BioLegend #MMS-435P). Images were taken using a Zeiss Observer.Z1 fluorescence microscope (Zeiss, Jena, Germany). Twenty- or forty-fold objectives and emission at 488 nm, 546 nm, 647 nm and 461 nm were used, depending on the respective secondary antibody. Analysis was performed with the FIJI image processing package by ImageJ (v1.53c).

### 2.4. Live Cell Imaging and Organelle Tracking Analysis

To analyze trafficking of the cellular organelles, motor neurons were stained with live cell dyes. Then, 50 nM LysoTracker Red dye, 50 nM MitoTracker Deep Red dyes or 100 nM JC-1 dye, which change fluorescence emission depending on the mitochondria’s membrane potential, were applied for at least 1 h at 37 °C and 5% CO_2_. Live imaging microscopy was performed with a 100× oil objective on a Leica Fluorescence microscope AF6000 TIRF by Andor (Leica, Wetzlar, Germany), using an iXON 897 EMCCD (Oxford instruments, Abington, UK) camera in a humidified incubation cabin at 37 °C and 5% CO_2_. For LysoTracker Red and MitoTracker Deep Red, images of both organelle trackers were acquired simultaneously over 2 min at 3 frames per second per channel. For JC-1, only single images were taken to analyze its staining in the red and green channel. Images were taken at an exposure time of 115 ms. The following settings were used: LysoTracker Red—excitation 561 nm, emission filter TRITC 605/65 nm, Mito-Tracker Deep Red—excitation 633 nm, emission filter Cy5 720/60 nm and JC-1—excitation 488 nm, and emission filters TRITC 605/65 nm and FITC 527/30. In MFCs, images were taken in the microchannels’ proximal entry and distal exit regions at strictly standardized readout positions.

Movies of axonal organelle trafficking in MFC microchannels were subjected to a quantitative tracking analysis using FIJI software (by ImageJ), as described in [53,54]. In brief, organelle tracking was performed with the TrackMate plugin. The following settings were used: pixel width: 0.230 µm, pixel height: 0.230 µm, voxel depth: 1.000 µm, time interval: 0.300 s, no cropping, DoG detector, estimated blob diameter: 1.6 µm for mitochondria, 1.6 µm for lysosomes, no median filtering, sub-pixel localization enabled, linear motion LAP tracker, initial search radius: 2.0 µm, search radius: 2.0 µm, max. frame gap: 2 frames, and duration of track filter: ≥3.00 s. Data were assembled, mined, and filtered with KNIME software (v3.5.2) for statistical analysis. Mobile lysosomes and mitochondria were defined by a track displacement ≥1.2 µm. JC-1 images were analyzed by measuring mitochondria object intensity with a size of more than 5 pixels in both the green and red channel after background subtraction. The intensity ratio of red/green channels was calculated to obtain a relative measure for the mitochondrial inner membrane potential.

### 2.5. Tracking Analysis

Movies were analyzed with FIJI software using the TrackMate v2.7.4 plugin for object detection and tracking (lysosomes and mitochondria). The settings were as follows: pixel width: 0.23 µm, pixel height: 0.23 µm, voxel depth: 1 µm, clipping settings: not applied, selected detector: DoG detector with an estimated blob size of 1.6 µm, threshold: 45, median filter: no, subpixel localization: yes, initial thresholding: none, select view: HyperStack displayer, filter set to spots with quality above 45, select tracker: linear motion lap tracker, initial search radius: 2 µm, search radius: 2 µm, max. frame gap: 2, and a set filter on tracks with a track duration of ≥3 s. Typically, 200–500 tracks per movie were obtained and analyzed for track displacement and mean velocity. The results were compiled and post-filtered in KNIME v3.5.2 and MS Excel v1611 (with a threshold for track displacement of ≥1.2 µm). The mass statistics were analyzed in GraphPad Prism 5 software and displayed as boxplots.

### 2.6. Static Analysis of Cell Organelles

The analysis of organelles count and morphology object segmentation, thresholding, and shape analysis was performed with a sequence of commands in FIJI software executed as described before [53]:run(“Slice Keeper”, “first=1 last=1 increment=1”);run(“Grays”);run(“Subtract Background…”, “rolling=3”);setAutoThreshold(“IsoData dark”);//run(“Threshold…”);run(“Convert to Mask”);run(“Set Measurements…”, “area fit shape feret’s redirect=None decimal=5”);run(“Analyze Particles…”, “size=4-Infinity pixel circularity=0.00–1.00 show=Ellipses display summarize”);and Marco2 for lysosomes:run(“Slice Keeper”, “first=1 last=1 increment=1”);run(“Grays”);run(“Enhance Contrast…”, “saturated=0.1 normalize”);run(“Subtract Background…”, “rolling=5”);setAutoThreshold(“Yen dark”);run(“Convert to Mask”);run(“Set Measurements…”, “area fit shape feret’s redirect=None decimal=5”);run(“Analyze Particles…”, “size=3-Infinity pixel circularity=0.40–1.00 show=Ellipses display summarize”);

These macros provided result tables with the aspect ratio of the fitted eclipses (long–short radius), which was used as a measure of mitochondrial elongation, and the outer Feret’s diameter, which was used as the lysosomal diameter. In general, hundreds of organelles were analyzed per film.

For analysis of mitochondrial membrane potential (ratio JC-1 of red–green channels), object segmentation was performed with the channel of higher intensity (mostly the red) to generate a selection limited to mitochondria using the following:resetMinAndMax();title=getTitle();run(“Slice Keeper”, “first=1 last=1 increment=1”);run(“Subtract Background…”, “rolling=10”);setAutoThreshold(“Default dark”);//run(“Threshold…”);run(“Convert to Mask”);run(“Create Selection”);

The resulting selection was saved as a region of interest (ROI) and applied to both channels to determine the total integral intensity and area of the mitochondria and background in both channels using the “Measure” command. After area normalization and background subtraction, the ratios of red–green integral intensity were taken as the mean membrane potential per movie (first frame only) and analyzed in the stack.

### 2.7. DNA Isolation and PCR

DNA was isolated by using an AllPrep DNA/RNA/Protein Kit (80004, Qiagen, Hilden, Germany) according to the manufacturer’s instructions. PCR was performed using a 2X Taq-polymerase mastermix (Thermo Fisher Scientific), 50 pmol of forward and reverse primer, and 100 ng of DNA under the following conditions: denaturation at 95 °C for 5 min; annealing at 95 °C for 30 s, T(m)-3 °C for 30 s, and 72 °C for 30 s (35 cycles); and elongation at 72 °C for 5 min. To quantify TAU isoforms, we used the following primer: totalMAPT-Fwd CTCGCATGGTCAGTAAAAGCAA; totalMAPT-Rev GGGTTTTTGCTGGAATCCTGGT; Exon10B-Fwd GTCAAGTCCAAGATCGGCTC; Exon10B-Rev TGGTCTGTCTGGCTTTGGC; MAPT3R-Fwd CAAAGACACCACCCAGCTCT; MAPT3R-Rev AGACTATTTGCACCTTCCCGC; TAU4R-Fwd CCAAGTGTGGCTCAAAGGAT; MAPT4R-Rev CCCAATCTTCGACTGGACTC.

### 2.8. Statistics

Column diagrams, box plots and their statistics were generated in GraphPad PRISM 7. Growth curves’ generation, Student’s *t*-tests, and SEM/SDTEV calculations were performed in Microsoft Office Excel v1611. A minimum of three independent experiments based on three different differentiation pipelines were always performed. If not otherwise stated, a one-way ANOVA was used for all experiments with a post hoc Bonferroni post test to determine statistical differences between groups. * *p* < 0.05, ** *p* < 0.01, *** *p* < 0.001, **** *p* < 0.0001 were considered significant. Data values represent mean ± SDTEV unless indicated otherwise.

## 3. Results

### 3.1. N297K but Not V337M Mutations in MAPT Led to a Pathological Increase in 4R TAU in FTDP-17 Patient-Specific Motor Neurons

We initially wanted to know whether mutations in MAPT affect neuronal differentiation or TAU isoform expression in FTDP-17 patient-derived motor neurons. To this end, we generated spinal MNs from FTDP-17 patient-specific NPCs carrying the mutations V337M or N279K as well as age- and gender-matched healthy controls, wild type 1 (WT1) and wild type 2 (WT2) (Figure 1A–C). Both controls, as expected, only showed 3R TAU expression. While V337M sMNs displayed only 3R TAU, N279K mutation shifted to 3R/4R TAU (Figure 1D). Furthermore, mature motor neurons displayed neuronal (TUJ1, MAP2) and motor neuronal (SMI32) marker expression (Figure 1E,F). These results correlate with the findings of previous studies in dopaminergic neurons [33].

### 3.2. Mitochondria and Lysosomal Trafficking Are Affected by V337K and N297K Mutations in MAPT of FTDP-17 Patient-Specific Neurons

TAU is predominantly expressed by neurons and particularly highly expressed in axons [52], and MAPT mutations were reported to differentially affect microtubule binding [29]. We thus chose two paradigmatic FTD-causing MAPT mutations with either increased (V337M) or decreased (N297K) microtubule binding and hypothesized that (I) MAPT mutations affect axonal trafficking of mitochondria and lysosomes and (II) that different TAU binding affinities differentially affect axonal trafficking of organelles in human patient-derived neurons. First, we investigated the trafficking of mitochondria and lysosomes in general in undirected cell culture by seeding 1 × 10^4^ to 4 × 10^4^ cells per 96 wells. Mitochondria (Figure 2A–G) and lysosomes (Figure 2H–M) were analyzed by acquiring movies of stained live organelles at three frames per sec over 2 min. Their motility was calculated by the assumption that immobile mitochondria/lysosomes move less than 1.2 µm in their track displacement. For mitochondria, there were no changes found in the mitochondrial size (Figure 2B) or in mitochondrial network size (as an approximation of overall mitochondrial mass) (Figure 2C). The ratio of high (hp) to low (lp) mitochondrial membrane potential measured by JC1 immunofluorescence live cell imaging was indistinguishable from controls in case of N297K cells but decreased in V337M cells (Figure 2D). Furthermore, the fraction of motile mitochondria of V337M cells was similar to WT1. However, the fraction of motile mitochondria in N297K cells was reduced by 14% as compared to WT2 (Figure 2E). Moreover, mitochondrial mean speed did vary in both mutant cell lines in an opposing manner. The mean speed of V337M mitochondria was decreased, whereas N297K cells displayed an increase (Figure 2F). Conversely, the track displacement of mitochondria was only decreased in N297K cells in comparison to WT2 (Figure 2G) but not in V337M versus WT1. As for lysosomes, we found no changes in size (Figure 2I) as well as in lysosomal network size (Figure 2J). The fraction of motile lysosomes in V337M cells was increased by 8% in comparison to WT1 as opposed to a decrease by 19% in N297K compared to WT2 (Figure 2K). The lysosomal mean speed was increased in both FTDP-17-patient specific cell lines (Figure 2L). In contrast, lysosomal track displacement in N297K cells was decreased, whereas V337M cells showed an increased displacement in comparison to WT1 or WT2 (Figure 2M).

### 3.3. The V337M MAPT Mutant Displays Severe Organelle Trafficking Defects in Axons

Next, we wanted to have a closer look at organelle trafficking defects of mitochondria and lysosomes in FTDP-17 mutations. To obtain data about organelle trafficking in specific regions and directions in sMNs, differentiated cells were seeded into microfluidic chambers (MFCs) (Figure 3A). An approximate axon growth rate was measured for 18 days after seeding by counting the microchannels that were already fully penetrated by protruding axons at a given time point. Penetrated channels were counted every two days from application of the growth factor gradient on day 2 onwards. It was found that both mutants and WT2 have the same growth rate, whereas WT1 growth significantly slower. Furthermore, V377K neuronal axons grew through all 120 channels within 10 days, while N297K axons did not grow through all of them until day 18 (Figure 3B).

Axonal mitochondria were analyzed by live cell imaging of labelled organelles at the proximal versus distal MFC channel ends (Figure 3C). We observed no changes in mitochondrial size (Figure 3D) as well as in mitochondria network size (Figure 3E) across all lines. Contrary effects were shown in the ratio of mitochondrial membrane potential in FTD cells. The ratio between high (hp) and low (lp) membrane potential was decreased in V337K cells at both sites (proximal and distal) in comparison to WT1 mitochondria, thereby matching the observations in the undirected cultures (compare Figure 3G and 2D). The control had a 1 hp/lp ratio, and the distal site’s ratio was similar to the control (Figure 3G). Moreover, the fraction of motile mitochondria was reduced by 13–15% in V337K cells at both sites (proximal and distal) compared to WT1, with a slightly more profound reduction on the distal site. Conversely, WT2 and N297K mutants were indistinguishable in their fraction of motile mitochondria of 17–20%, which was a bit lower than the fraction of WT1 (Figure 3H). In V337K cells, mitochondrial mean speed was drastically decreased in both the proximal and distal site compared to WT1, whereas N297K cells showed no appreciable difference to WT2 apart from a slight increase only at the distal site (Figure 3I). In analyzing the mitochondrial track displacement in V337K cells, we revealed that organelles in the proximal and distal region were transported over a drastically reduced net distance in comparison to the controls, consistent with their decreased mean speed and hence a loss of straight, efficient transport. By contrast, N297K cells displayed a slight increase in mitochondrial track displacement compared to WT2 (Figure 3J). Additionally, there were an upregulation in the proximal side of N279K lysosome aspect ratio in comparison to WT2 (Figure 3K). Subsequently, lysosome trafficking was analyzed, and it was found that the motility in V337K cells as well as in N297K cell lysosomes did not vary from the WT (Figure 3L). The lysosome mean speed was decreased in V337K cells compared to WT1, with a more profound reduction at the distal site; however, in N297K cells, this reduction was much more moderate (Figure 3M). The organelle track displacement was decreased in the proximal region of both mutants in comparison to WT1 or WT2 (Figure 3N). Nevertheless, there was no change in the network size of lysosomes in FTDP-17 cells (Figure 3O).

In summary, axonal mitochondrial and lysosomal trafficking in MNs were profoundly affected by the V337M mutation, whereas the N297K mutation had a much more subtle impact.

## 4. Discussion

The prevalence of neurodegenerative diseases is increasing yearly and the treatment of these diseases poses major challenges to the healthcare system. Most if not all of these diseases cannot be efficiently treated yet. Thus, there is an urgent need for the development of novel therapeutic strategies. Possible factors inducing neuronal death were found to correlate with defects in organelle trafficking along neurites. Particularly in long, vulnerable axons, misregulation of organelle trafficking poses a higher risk of detrimental supply failures, thereby leading to cellular demise. Mitochondrial and lysosomal trafficking are essential processes within the cell, and defects in these organelles can cause alterations in (subcellular) metabolism and autophagy putatively leading to protein aggregation and finally to neurodegenerative pathologies [45]. TAU is mainly expressed in neurons, particularly in axons [52], marking this protein a valuable candidate for explaining selective susceptibility to neurodegeneration. MAPT mutations have been reported to differentially affect microtubule binding [29], suggesting that this leads to different axonopathies. FTDP-17 is caused by MAPT mutation, e.g., V337M (resulting in a decreased interaction of TAU protein with microtubules) or N297K (resulting in increased association of TAU protein with microtubules). To this end, iPSC-derived patient-specific sMNs with these two paradigmatic MAPT mutations (V337M, N297K) were generated and investigated in undirected and directed neuronal cell culture in this study regarding organelle trafficking (Figure 4).

Firstly, there was no impairment of overall neuronal differentiation found in FTDP-17 NPCs. Spinal MNs with FTDP-17-causing mutations, i.e., V337M or N297K, displayed no differences in axonal outgrowth in comparison to WT2 as well as altered axonal trafficking of mitochondria and lysosomes.

Observing the axonal outgrowth of FTDP-17 patient-specific sMNs over a period of 18 days in directed cell culture, mutants and WT2 grew in a similar manner, but differed to WT1. We thus assume that the differences in growth are not due to the mutations but rather due to the genetic background or other influences. Comparing our results with neurite outgrowth in midbrain dopaminergic neurons, V337M as well as N279K MAPT cell neurites were shown to grow significantly slower than control cell ones within 48 h [33]. Reduced axonal outgrowth of N279K MAPT-carrying cells has also been previously reported in cortical neurons [55]. A reason for these discrepancies could be (I) the different types of analyzed neurons as well as (II) methodological differences. Ehrlich and colleagues analyzed cortical and midbrain neurons in an undirected cell culture over just 48 h, whereas we analyzed spinal neurons in MFCs over 18 days with long microchannels of 900 µm in length.

The N279K MAPT mutation leads to increased microtubule stability, reduced polymerization, and increased fibril formation due to a ratio shift of TAU isoforms towards 4R ones [29]. Even though it is reported that iPSC-derived neurons lack TAU isoform expression other than 3R due to their fetal character [56], we and others [33] could show that N297K neurons expressed 3R and 4R, which was not found in neither controls or the V337M mutant. In undirected cell culture, decreased motility and displacement of mitochondria and lysosomes were shown alongside an increase in the mean speed in both of them. The reduced displacement despite increased speed could be assumed to be trafficking with several breaks. Similar to our results, the appearance of an increased amount of stationary mitochondria was previously reported by Iovino et al. [57]. Using directed cell culture, we observed an increase in mitochondrial membrane potential, mean speed, and displacement in distal axons. Increased membrane potential in mitochondria alongside a spontaneous calcium oscillation leading to mitochondrial hyperphosphorylation and ROS production has been recently reported in cortical neurons carrying the N297K mutation. Furthermore, Korn et al. were able to show that specific proteome clusters regulating apoptosis, ROS homeostasis, and mitochondrial function are primarily misregulated by the N297K mutation [58]. Together with our data, this shows that N297K mutation leads to defective mitochondria in different types of neurons. How this is connected to the increased occurrence of the 4R tau in this MAPT variant needs to be investigated. Further studies should also clarify the direct cellular consequences of potential alterations of the mitochondrial membrane.

Undirected MN cultures revealed that organelles were contrarily affected in V337M MAPT carrying cells. Mitochondria were found to be decreased in their mean speed as well as in their membrane potential, whereas the amount of motile organelles and their displacement were similar to control ones. However, lysosomal trafficking was increased in speed, displacement, and amount of moving organelles. In directed cell culture the results of mitochondria analysis correlated with the undirected cell culture. In contrast, the comparison of lysosomes in undirected and directed cell culture was very different. Whereas in undirected cell culture, the mean speed and the displacement were upregulated, they were downregulated in directed cell culture. These differences might arise from the fact that in directed cultures in microfluidic chambers, axons are mainly analyzed, while in undirected cultures, both axons and dendrites cannot be distinguished; thus, our results represented a mixture of both. This is of note because lysosomes are more abundant in dendrites than in axons [48,59]. Consequently, both mutations mainly affect the mean speed and displacement of mitochondria and lysosomes, whereas the morphology is largely intact. Therefore, it can be assumed that both the V337M and N297K mutations lead to a deterioration in microtubule stability and thus hinder the transport of cellular organelles such as mitochondria and lysosomes. The V337M mutation only expresses 3R TAU and thus leads to the reduced binding and lower stability of microtubules [11,26,29]. The N297K mutation is more effective in building microtubules due to the increased formation of 4R TAU and thus does not appear to affect the binding of TAU to microtubules. On the other hand, this mutation is known for increased TAU aggregate formation, which could also destabilize microtubules through increased binding of aggregated 4R TAU [30,31]. Nevertheless, there could also be reduced binding of TAU in both mutations to microtubules due to hyperphosphorylation. However, it has also been shown that neuronal stem cells carrying the N279K mutation exhibit higher cellular stress and display a disrupted intracellular vesicle trafficking [36]. Additionally, this work showed that N279K neuronal stem cells possessed a marked depletion of lysosomal compartments and protein levels of lysosomal marker LAMP-1, which supports our findings of impaired lysosome transport.

In our study, we used well-characterized iPSC-derived functional motor neurons. However, their in vitro cultivation differs from the situation in vivo (e.g., much shorter axons, etc.). In addition, the dyes used in our experimental setup have some limitations. LysoTracker dyes stain lysosomes due to their low pH; thus, they label lysosomes only as long as their pH is beneath a certain threshold. Furthermore, late endosomes also have a similar pH [60,61]. On the other hand, staining of mitochondria with MitoTracker is to some extent dependent on their membrane potential, so only organelles with high potential are visualized [62]. We depicted a difference in the mitochondrial potential of these mutants; thus, this could be also a measure of the phenotype in the respective mutations. Nevertheless, we cannot exclude that this masks some phenotypes in the presented setting. However, all live cell dyes share these properties to some extent. In addition, both LysoTracker and MitoTracker display low photostability [63].

## 5. Conclusions

Our results show that iPSC-derived sMNs carrying the V337M or the N297K MAPT mutation displayed defects in the transport of membrane-bound organelles (mitochondria and lysosomes), presumably due to destabilization of microtubules due to mutated TAU protein. These results have important implications for our understanding of the pathogenic mechanism of the N279K and V337M TAU mutation as well as for FTD pathology and may be partially consistent with sporadic FTD. In addition, our results emphasize that new therapies will probably have to be developed on a mutation-specific basis or depending on changes in 3R/4R induced by respective mutations. Further studies are required to elucidate the precise pathways by which the MAPT mutations studied here alter organelle trafficking.

## Figures and Tables

**Figure 1 biomedicines-12-00641-f001:**
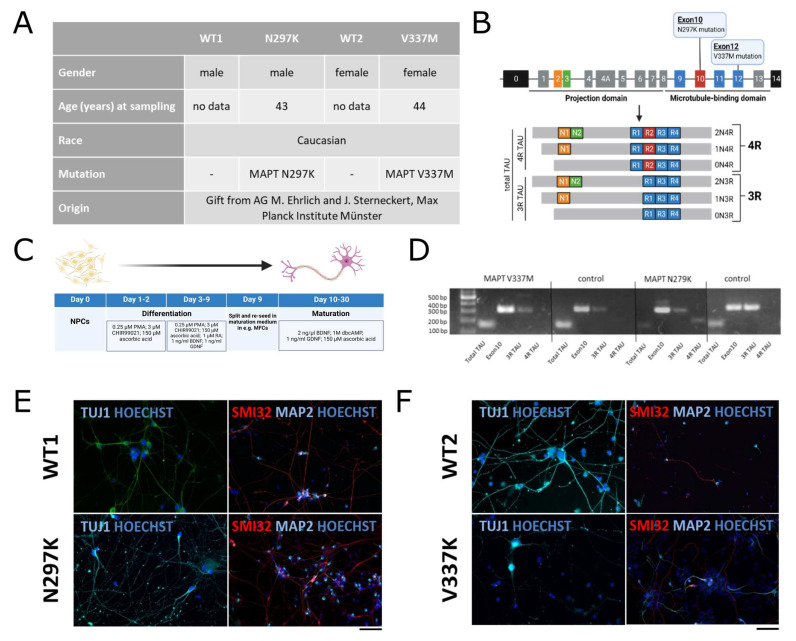
Differentiation and characterization of patient-derived iPSC lines used in this study. (**A**) Overview of patient characteristics of derived cell lines. (**B**) Localization of N297K and V337 mutations in the MAPT gene and TAU isoforms. (**C**) Differentiation protocol from neuronal pro-genitor cells (NPCs) to spinal motor neurons in 10 to 30 days. (**D**) Genome analysis of total and different TAU isoforms in all cell lines. (**E**,**F**) Immunofluorescent staining of neuronal (TUJ1, MAP2) and motor neuronal (SMI32) marker proteins. Scale bar = 50 µm.

**Figure 2 biomedicines-12-00641-f002:**
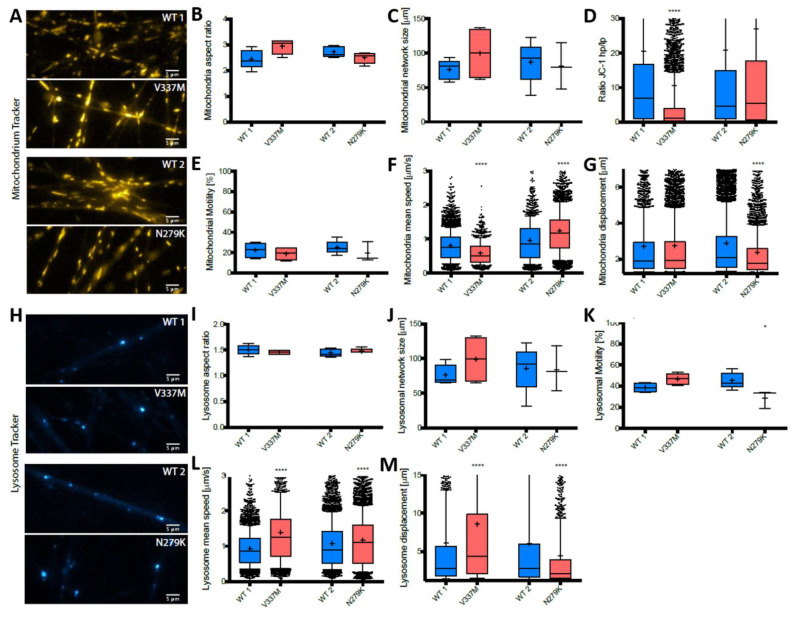
Analysis of mitochondria and lysosomes of V337M and N297K mutated sMNs in undirected cell culture. (**A**,**H**) Representative pictures of MitoTracker (**A**) and LysoTracker (**H**) live cell imaging in V337M and N297K as well as control cells in undirected cell culture. Scale bar: 5 µm. (**B***–***G**) Box plot analysis of mitochondria shape and aspect ratio (**B**), mitochondrial network size (**C**), and membrane potential (**D**) as well as mitochondrial motility (**E**), mean speed (**F**), and displacement (**G**). (**I**–**M**) Lysosome tracking box plot analysis of size and aspect ratio (**I**) and lysosomal network size (**J**) as well as lysosomal motility (**K**), mean speed (**L**), and displacement (**M**) [one-way ANOVA with post hoc Bonferroni post test; * *p* < 0.05, **** *p* < 0.0001; data values represent mean ± SDTEV]. Blue bars depict controls, red bars mutants.

**Figure 3 biomedicines-12-00641-f003:**
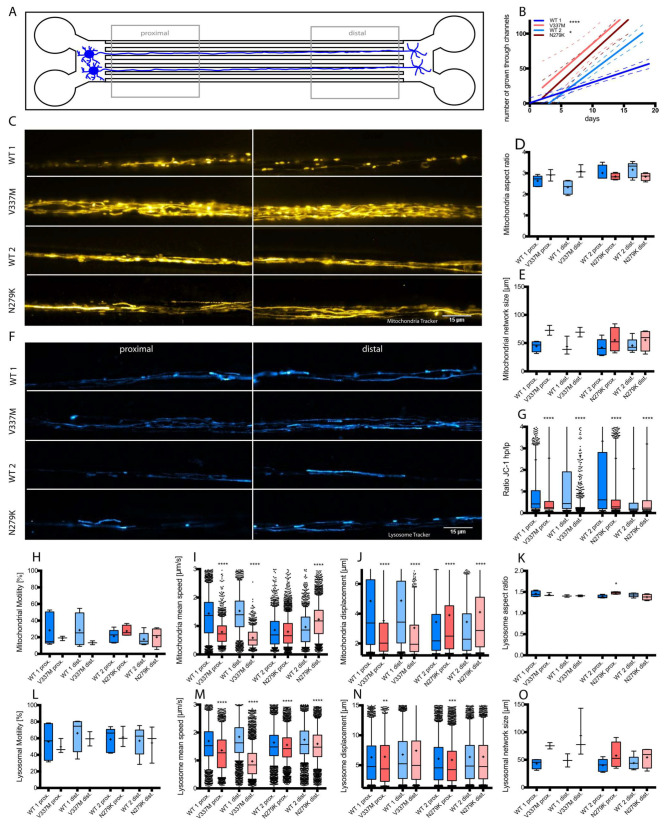
Analysis of mitochondria and lysosomes of V337M and N297K mutated sMNs in directed cell culture. (**A**) sMNs were differentiated and seeded into MFCs. Directed growth of axons through 900 µm long microchannels was realized by application of a volume and maturation factor gradient functioning as a guidance cue. For standardized statistical readout, “distal” and “proximal” windows were defined. (**B**) Axonal growth curve of sMNs in MFC cell culture. The number of channels which were passed through by axons were counted every 2 days for 18 days. The total amount of channels is 120. (Student’s *t*-test * *p* < 0.05, **** *p* < 0.0001; data values represent mean ± SDTEV]. (**C**,**F**) Representative pictures of MitoTracker (**C**) and Lysosome Tracker (**F**) live cell imaging in V337M and N297K as well as control cells in MFC cell culture. Representative pictures of the proximal and distal part of the sMNs are shown. Scale bar: 15 µm. (**D**,**E**) Organelle tracking analysis as box plots of mitochondria size (**D**) and network size (**E**) [one-way ANOVA with post hoc Bonferroni post test; data values represent mean ± SDTEV]. (**G**–**J**) Box plot analysis of mitochondrial membrane potential (**G**), motility (**H**), mean speed (**I**), and displacement (**J**) [one-way ANOVA with post hoc Bonferroni post test; **** *p* < 0.0001; data values represent mean ± SDTEV]. (**K**–**O**) Box plot analysis of lysosomal size (**K**), motility (**L**), mean speed (**M**), displacement (**N**), and network size (**O**) [one-way ANOVA with post hoc Bonferroni post test; * *p* < 0.05, ** *p* < 0.01, *** *p* < 0.001, **** *p* < 0.0001; data values represent mean ± SDTEV]. Blue bars depict different conditions of the controls, red bars those of mutants.

**Figure 4 biomedicines-12-00641-f004:**
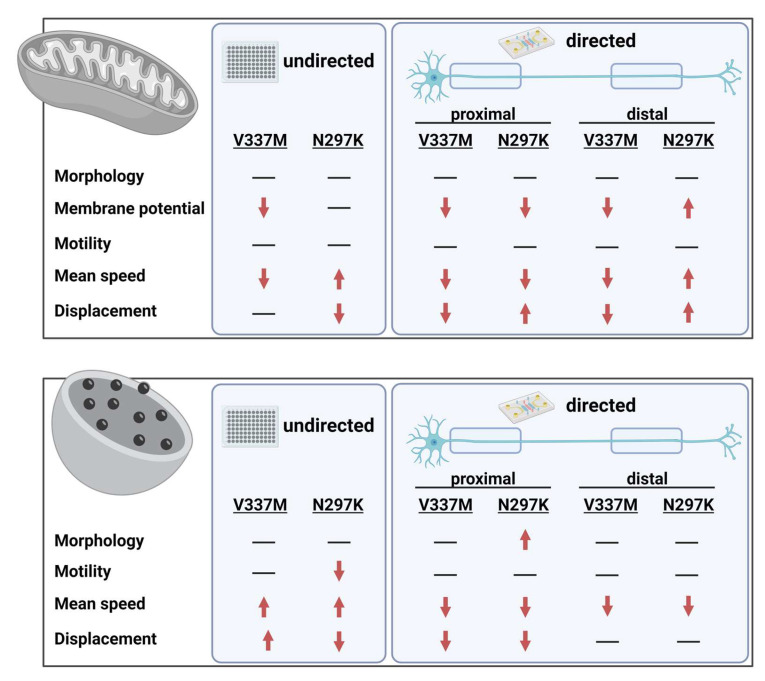
Summary of organelle trafficking analysis. Mitochondria and lysosomes were analyzed in undirected and directed (MFCs) cell culture to investigate alterations in organelle trafficking. The results of our analyses are summarized in this graphical abstract.

## Data Availability

All data are presented in this manuscript.

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
