# Peer review of "MAPT Mutations V337M and N297K Alter Organelle Trafficking in Frontotemporal Dementia Patient-Specific Motor Neurons"

_biomedicines, 2024, doi:10.3390/biomedicines12030641_

Round 1

Reviewer 1 Report

Comments and Suggestions for Authors

Reviewer comments

Microtubule-associated protein TAU (MAPT) is one of the major proteins associated with neuropathology, and defects in this protein lead to neurodegenerative disorders, including dementia. In the present research, the article authors have observed the two mutations V337M and N297K in tau protein that affect microtubule stabilization and organelle transportation, specifically mitochondrial and lysosomal trafficking. Both mutations cause an increase in axonal outgrowth and impaired axon physiology. Therefore, it concluded that this tau protein and its associated mutation could be a therapeutic target for dementia treatment and management.

Recommendation

1.      The paper is scientifically sound, well-planned, and written in an organized manner.

2.      The research objectives and targets of the work are original and relevant to the field. The outcomes of the study are discussed nicely.

3.      Experimental result figures are correctly explained in the result text and scientifically discussed in the discussion section.

4.      References are appropriate.

Minor comments/suggestions

1.      Discuss this paper: doi: 10.1016/j.redox.2022.102597

2.      How this study will help in the development of a new therapeutic approach in the future should be discussed briefly.

3.      Use the same pattern of writing for the word 'Figure'.

4.      Line 228, check the citation pattern for Naumann et al. (2018).

5.      Check for the way of writing for power digits, eg in lines 280-1x104 to 4x104 and at other places.

Author Response

Reviewer 1

Recommendation

  1. The paper is scientifically sound, well-planned, and written in an organized manner.
  2. The research objectives and targets of the work are original and relevant to the field. The outcomes of the study are discussed nicely.
  3. Experimental result figures are correctly explained in the result text and scientifically discussed in the discussion section.
  4. References are appropriate.

We deeply thank the reviewer for this very positive overall review.

Minor comments/suggestions

  1. Discuss this paper: doi: 10.1016/j.redox.2022.102597

We included the publication in our discussion.

  1. How this study will help in the development of a new therapeutic approach in the future should be discussed briefly.

We added a brief comment that our results highlight a mutation-specific therapeutic approach in the discussion.

  1. Use the same pattern of writing for the word 'Figure'.

Thanks to point out this. We now use the same spelling for the word “Figure” in the whole manuscript.

  1. Line 228, check the citation pattern for Naumann et al. (2018).

Thanks to point out this. We corrected this citation pattern.

  1. Check for the way of writing for power digits, eg in lines 280-1x104 to 4x104 and at other places.

Thanks to point out this. We corrected the writing for the power digits.

Reviewer 2 Report

Comments and Suggestions for Authors

This paper investigates organelle trafficking defects in Frontotemporal Dementia (FTD) patients carrying MAPT mutations (V337M, N297K) using live cell imaging. The study explores differential effects on axonal physiology, highlighting the need for mutation-specific therapeutic approaches. The paper is well-structured and informative. However, clarity in some sections and data interpretation could be improved.

  1. In the introduction, a broader context of FTD prevalence and impact could enhance the significance of the study. Can this be expanded upon?
  2. The study notes altered mitochondrial membrane potential. How does this correlate with other mitochondrial dysfunctions observed in FTD?
  3. Can you provide more details on the methods employed for live cell imaging and data analysis to enhance reproducibility?
  4. The paper mentions increased axonal outgrowth in both mutants. What implications does this have for the pathophysiology of FTD?

Could the paper discuss potential limitations and challenges faced during the investigation of organelle trafficking in patient-derived neurons?

Author Response

Reviewer 2

This paper investigates organelle trafficking defects in Frontotemporal Dementia (FTD) patients carrying MAPT mutations (V337M, N297K) using live cell imaging. The study explores differential effects on axonal physiology, highlighting the need for mutation-specific therapeutic approaches. The paper is well-structured and informative. However, clarity in some sections and data interpretation could be improved.

We deeply thank the reviewer for this very positive overall review and addressed all remaining comments (below).

  1. In the introduction, a broader context of FTD prevalence and impact could enhance the significance of the study. Can this be expanded upon?

Data on FTD prevalence was added to support the scope of the study

  1. The study notes altered mitochondrial membrane potential. How does this correlate with other mitochondrial dysfunctions observed in FTD?

We discussed the altered mitochondrial potential with other studies. Together with our data, it is likely that an increased occurrence of the 4R tau variant leads to an increase in oxidative stress that might lead to cell death.

  1. Can you provide more details on the methods employed for live cell imaging and data analysis to enhance reproducibility?

We described the live cell imaging in more detail and add additional information about the Macro we are used.

  1. The paper mentions increased axonal outgrowth in both mutants. What implications does this have for the pathophysiology of FTD?

We found that both mutants and WT2 have the same growth rate whereby WT1 growth significantly slower. Because WT2 display also an increased growth rate it could be assumed that the differences in growth are not due to the mutations, but to the genetic background or other influences. We mentioned this in the discussion.

  1. Could the paper discuss potential limitations and challenges faced during the investigation of organelle trafficking in patient-derived neurons?

We added a section about the limitation of our cell model and the used fluorophores in the discussion. Both, LysoTracker and Mitotracker are photosensitive and might label not all organelles, because the staining depends either on the pH of the lysosomes or the membrane potential of mitochondria.